# Co-Crystallization Approach to Enhance the Stability of Moisture-Sensitive Drugs

**DOI:** 10.3390/pharmaceutics15010189

**Published:** 2023-01-05

**Authors:** Madhukiran R. Dhondale, Pradip Thakor, Amritha G. Nambiar, Maan Singh, Ashish K. Agrawal, Nalini R. Shastri, Dinesh Kumar

**Affiliations:** 1Department of Pharmaceutical Engineering and Technology, Indian Institute of Technology (BHU), Varanasi 221005, India; 2Natco Research Center, Natco Pharma Limited, Hyderabad 500018, India; 3Solid State Pharmaceutical Research, Hyderabad 500037, India

**Keywords:** co-crystals, stability, moisture-sensitive, hygroscopicity

## Abstract

Stability is an essential quality attribute of any pharmaceutical formulation. Poor stability can change the color and physical appearance of a drug, directly impacting the patient’s perception. Unstable drug products may also face loss of active pharmaceutical ingredients (APIs) and degradation, making the medicine ineffective and toxic. Moisture content is known to be the leading cause of the degradation of nearly 50% of medicinal products, leading to impurities in solid dose formulations. The polarity of the atoms in an API and the surface chemistry of API particles majorly influence the affinity towards water molecules. Moisture induces chemical reactions, including free water that has also been identified as an important factor in determining drug product stability. Among the various approaches, crystal engineering and specifically co-crystals, have a proven ability to increase the stability of moisture-sensitive APIs. Other approaches, such as changing the salt form, can lead to solubility issues, thus making the co-crystal approach more suited to enhancing hygroscopic stability. There are many reported studies where co-crystals have exhibited reduced hygroscopicity compared to pure API, thereby improving the product’s stability. In this review, the authors focus on recent updates and trends in these studies related to improving the hygroscopic stability of compounds, discuss the reasons behind the enhanced stability, and briefly discuss the screening of co-formers for moisture-sensitive drugs.

## 1. Introduction

The hygroscopic character of drugs or excipients may induce unexpected phase transformations and instability issues, which affect the performance of the material during various stages of the pharmaceutical process, such as manufacturing, packing, storage, and transport. This also has a pronounced effect on the appearance and effectiveness of the final product. Hygroscopicity is a phenomenon in which atmospheric water vapor is taken up by a molecule and retained through non-covalent interactions, especially hydrogen bonding, at a given temperature and relative humidity (RH) [1]. A material that ‘absorbs’ or ‘adsorbs’ moisture from the surroundings undergoes changes in its physical properties, such as volume, melting point, etc. This moisture absorption/adsorption also affects the product characteristics, such as flowability, compressibility, sticking and picking, and the processability of the material, such as mixing, coating, drying, etc. [2]. The adsorption phenomenon has a significant impact on the surface properties of solids without affecting bulk properties. For example, on the macroscopic properties of solids, such as particle conglomeration and flow with a very high specific surface area; this is because water adsorption can alter the surface energy and affect the stability and shelf life of moisture-sensitive drugs [3].

Polymorphic transitions of API, like amorphous to crystalline forms, anhydrate to hydrate formations, color changes, and other visual changes like swelling, capping, or twinning of tablets, are also induced by moisture [4]. Moisture can react with crystalline solids and result in deliquescence, crystal hydrate formation, and the adsorption of water vapor into the solid-air interface [5]. At low RH, capillary condensation can occur in solids having micro void spaces [2]. Hydrate formation takes place when the water molecules occupy well-defined positions within the unit cells of a crystal lattice. Furthermore, the water molecules form hydrogen-bonded monolayers at the surface of the solid [2,6]. The affinity of water molecules and subsequent adsorption further depends on the functional groups that are present on the surface of the API particles. Based on the type of functional groups of a molecule that are exposed to the environment, the moisture ingress into the particles occurs at different proportions [7,8]. This adsorption is quite easily reversed by the decrease in RH or by the increase in temperature. Irregularities in crystalline solids increase the free energy, resulting in a higher amount of moisture uptake than in a pure crystalline state [2,9].

Amorphous materials undergo a plasticization effect after moisture uptake from the environment due to a decrease in glass transition temperature (T_g_) which decides the stability of a particular amorphous substance. The changes in T_g_ cause the amorphous materials to undergo a phase transition from a “glassy” to a “rubbery” state, as well as increasing the molecular mobility, leading to significant changes in the free volume and viscosity [10]. Ultimately, moisture can affect various physicochemical attributes of pharmaceutical concern involving amorphous and crystalline solids, e.g., induction of crystal growth in lyophilized cakes, direct compression properties, powder caking, permeability properties of coatings and packaging materials, and solid-state chemical stability [9].

Various control strategies have been employed to improve the hygroscopic stability of susceptible APIs (Figure 1), such as the use of manufacturing controls [11], protective packaging of dosage forms [12], coating with hydrophobic polymers/lipids [13], encapsulation in lipid formulations [14], changing salt forms [15], formation of co-crystals [16], etc. The advantages and disadvantages of each control strategy are outlined in Table 1. It has been noted from recent publications that co-crystallization has been used as an effective approach to enhance the stability of moisture-sensitive APIs [16,17,18]. Among these methods, the crystal engineering approaches include changing salt forms and the preparation of co-crystals. The purpose of preparing salts is to improve the solubility of APIs. If any changes to the salt form are made, the solubility of that particular compound is also compromised [19].

Generally, solid pharmaceutical dosage forms show better stability than liquid dosage forms. Furthermore, the use of co-crystals in place of plain APIs can add significant advantages to the stability of the solid dosage form. Co-crystals have received significant attention in the last two decades as a novel solid form because of their capability to modify the pharmaceutical and biopharmaceutical properties of APIs [24,25,26,27]. The co-crystals are formed by an API and co-formers with the action of non-covalent bonds. Co-crystals provide an effective approach to enhancing the stability of APIs in multiple ways; this is discussed in detail in a later section of this review article [28]. The use of substances from the USFDA list of generally recognized as safe (GRAS) nutraceuticals as co-formers makes the co-crystals devoid of any toxicity due to the use of co-formers [29]. Figure 2 shows the number of scientific publications from 2003 to 2022 published in Science Direct on the topics “drug stability, hygroscopicity” and “co-crystals, drug stability, hygroscopicity”. Co-crystallization of APIs to solve the stability/hygroscopicity issue was almost non-existent two decades ago. Due to their effectiveness in improving the stability of problematic APIs, co-crystals have gained notable interest from the research fraternity across the globe. Currently, almost 50 literature works have been published on the topic of co-crystals for improving the hygroscopic stability of APIs. Also, a significant number of publications have shown the research and development of co-crystals and their applications to improve aqueous solubility, dissolution rate, manufacturability, and the oral bioavailability of APIs (Appendix A). The research and development of co-crystals is also supported by the updated Food and Drug Administration (FDA) guidelines, which consider co-crystals and solvates as drug polymorphs, not new APIs. The FDA defines co-crystals as “Crystalline materials composed of two or more different molecules, one of which is the API, in a defined stoichiometric ratio within the same crystal lattice that is associated by non-ionic and noncovalent bonds”. Accordingly, co-crystals represent a separate class of compounds and differ from solvates, salt forms, co-amorphous mixtures, etc. [30]. The European Medicinal Agency (EMA) also released guidelines for the quality control of co-crystals [31]. These regulatory inclusions motivated researchers and manufacturers to bring more pharmaceutical co-crystals into the market. This shift has been evident in the regulatory approval of co-crystals.

The possible structural and molecular mechanisms by which the co-crystals show reduced hygroscopicity is still under exploration. Various researchers have constantly been trying to undermine this particular aspect of co-crystals that enhances their hygroscopic stability. Also, the choice of co-former for a particular API molecule depends on the property under question that needs to be enhanced. In this paper, we have tried and reviewed the characteristics of API and co-former molecules that result in the reduction of their hygroscopic stability when prepared as co-crystals. The methods to assess the moisture content in a sample are briefly outlined in the last section of this review.

## 2. Moisture Interaction with Solid

### 2.1. Interaction Mechanism

The phenomenon when water interacts only at the surface of solids is termed adsorption, and when water penetrates the bulk solid structure, it is termed absorption. This moisture can either be adsorbed as mono/multilayers or exists as condensed moisture. The adsorbed water is also known as free water because it is loosely bound to the particle surface and behaves like pure water. Conversely, the absorbed water tightly bound to the particles is generally not available for eliciting a chemical reaction. Free water provides plasticity to the powder and is readily available for a chemical reaction. Thus, free water is more critical than bound water to the physicochemical stability of moisture-sensitive drugs and excipients [32].

The hygroscopic APIs mainly interact with the water molecules through intermolecular hydrogen bonding. The hygroscopic materials, such as berberine chloride [33], theophylline [34], palmatine chloride [35], oxymatrine [36], etc., have several hydrogen bond donors or acceptors in their chemical structure that are involved in the hydrogen bonding with the atmospheric moisture and thereby facilitate adsorption and subsequent ingress of water into the crystal lattice of these molecules. Depending upon the type of hydrogen bonding site (-OH, -COOH, -NH, -X, etc.) present in the molecule, different types of supramolecular synthons are formed with the water molecule.

Surface chemistry plays an important role in deciding the physical and chemical properties of a particular substance. The type of functional groups on the surface of a crystal can alter the solubility and hygroscopicity of a compound [37,38]. The presence of hydrophilic functional groups towards the surface of the crystal face is found to be responsible for the higher moisture uptake, as seen in a study by Watanabe et al. [8]. The study explored the correlation between the hygroscopicity of a compound and functional group orientation across different facets of the crystal. Four different co-crystals of isosorbide (ISO) with piperazine (PZ), hydrochlorthiazide (HCT), 3,5 dihydroxy benzoic acid (DHBA), and gallic acid (GA) were prepared, and the moisture uptake behavior at 30% RH of all the compounds was studied. The functional group type and their relative numbers present at the major crystals surface of the ISO and other co-crystals significantly influenced the water adsorption behavior at 30% RH (which is well below the critical RH of all the compounds under study). As seen in Table 2, the ISO-DBHA co-crystal showed the highest moisture uptake at 30% RH because of the internalization of DBHA, and as a result, the ISO and its hydroxyl groups were present on the surface of the crystal (Figure 3). This particular feature in ISO-DBHA co-crystals resulted in higher moisture adsorption relative to the other substances under study at 30% RH. This study essentially explained the reason for the surface adsorption of moisture by hygroscopic drugs.

### 2.2. Hygroscopicity of Hydrochloride Salts: Case of Berberine Chloride, Palmatine Chloride, and Adiphenine Hydrochloride

Poor water solubility is an often-encountered issue with APIs, and salt formation is the most common solution to counter the solubility issues. Hydrochloride/chloride salts, in particular, are widely used for the solubility enhancement of basic drugs [39]. Although the chloride salt form of a drug solves the solubility issues, it tends to cause problems with respect to the stability of that compound. It can be speculated that the electronegative atom chlorine of the drug’s salt form acts as a hydrogen bond acceptor and thereby facilitates hydrogen bonding with the atmospheric moisture. As a result, the compound becomes hygroscopic, thus causing major handling and stability issues and compromising the quality of the dosage form. The following examples support the aforesaid statement. Berberine chloride (BCl) and palmatine chloride are the major alkaloidal compounds used in herbal preparations [16,23,35,40]. The hydrochloride salt forms of both compounds are hygroscopic but their other salt forms and co-crystals are relatively less hygroscopic in nature. BCl at higher RH conditions forms hydrogen bonding with the water molecules and forms dihydrate and tetrahydrate forms. Whereas the salt co-crystal of berberine chloride with citric acid involves chlorine atoms in the hydrogen bonding network and prevents hydrogen bonding with water molecules [16]. Co-crystal hydrate of berberine chloride with L(+)-lactic acid also involves hydrogen bonding between the chlorine atom with the co-former and neighboring berberine chloride molecule [17]. The formation of berberine chloride and emodin co-crystals also involves chlorine atoms [40]. As a result of the involvement of chloride of API in hydrogen bonding with co-formers, the stability of BCl against hygroscopicity improves significantly (Table 3). Similarly, in the case of palmatine chloride salt, it can be assumed that the presence of chlorine atoms contributes to a greater extent to the hygroscopicity of the molecule. There has been an attempt to reduce the hygroscopicity of palmatine chloride by preparing a sulfosalicyate salt of the compound. There was around a 7% reduction in the weight gain of sulfosalicyate salt compared to the chloride salt of palmatine at 98% RH. Although the novel salt form exhibited low hygroscopicity, the solubility of the salt form was reduced [19]. A similar observation was noted with the palmatine saccharinate salt and gallic acid-palmatine chloride co-crystal, which showed better hygroscopic stability but reduced its aqueous solubility [23,35]. Another study by Ribeiro et al. [15] aimed at reducing the hygroscopicity of adiphenine hydrochloride by preparing more stable oxalate and citrate salt forms. This greatly enhanced the stability of the molecule but the aqueous solubility reduced significantly. Through these observations, it can be concluded that the hydrochloride/chloride salt forms are essential for the aqueous solubility of poorly soluble drugs but are also responsible for the hygroscopicity of drugs. Changing the salt form indeed reduces the hygroscopicity, but at the expense of reduced aqueous solubility. Figure 4 depicts the structures of BCl and its co-crystals, and palmatine salts and their co-crystals.

## 3. Impact of Moisture-Sensitive Materials on a Process and the Final Product

Moisture sorption by the dosage forms severely impacts its physical and chemical stability (Figure 5). The chemical stability of the API in the final dosage form is considered the most important concern. In hygroscopic materials, alternation of water content can affect the extent of degradation and has an indirect effect on the stability of products. This mainly affects the shelf life of the product [41].

### 3.1. Chemical Instability

There are various mechanisms by which the bound/absorbed water affects the chemical stability and consequently the performance of a pharmaceutical product.

(1) The water may degrade APIs by inducing hydrolytic reactions.

(2) It could be a product of specific reactions and may have an influence on product chemical reactivity and also may inhibit the forward reaction.

(3) The moisture in the amorphous matrix of the material could also induce reactions by varying the reactivity and polarity of the adjoining molecules [41].

Many of the excipients employed commercially are amorphous or semicrystalline hydrophilic polymers. On account of them containing polar functional groups, these materials have the ability to sorb large quantities of water and hence affect their functionalities [42]. As shown in Figure 6, moisture can negatively affect the T_g_ of compounds, which is a key player in the chemical stability of APIs and excipients and induces its degradation [43,44].

Moisture-sensitive drugs like aspirin can undergo hydrolysis and hence, having stringent control of the process parameters is required to ensure minimum exposure to water during the unit operations [45]. Chen et al. [46] reported the hydrolysis of lactone moiety of simvastatin in the presence of hygroscopic excipient sorbitol in tablets stored at high RH. Mihranyan et al. [47] studied the effect of the different grades of cellulose and its properties on the extent of degradation of aspirin induced by moisture. The crystallinity and moisture content of the excipients affected the degradation rate of aspirin to a great extent. The authors inferred that the region and availability of sorbed moisture were more important than the total water content of the system.

### 3.2. Physical Instability

Drug products must retain their physical quality and performance, in addition to chemical stability, at the end of their shelf life. The physical instability in formulation due to moisture is discussed briefly in the following section.

#### 3.2.1. Flow Property

The flow property of solids is affected by the amount of moisture (Figure 7A) [48,49]. The storage (particularly the RH) conditions decide the thickness of the adsorbed liquid layer on the particles and the strength of the liquid bridges formed between the particles and moisture. Consequently, at high RH, the cohesiveness of the solid is increased and leads to the formation of agglomerates [49,50]. Therefore, RH is a key factor that describes the boundary conditions affecting the moisture content of the powder [51].

Emery et al. [50] studied the flowability of moist pharmaceutical powders like HPMC (hydroxyl propyl methyl cellulose) and aspartame. They concluded that the flow property of aspartame was increased with an increase in moisture content up to 8%, which was associated with the formation of large and round agglomerates. However, above 8%, the flow property of aspartame decreased due to the increasing strength of liquid bridges that retarded the free flow of powder. HPMC with <5% moisture exhibited poor flow properties due to the increased cohesiveness. When moisture content increased beyond 5% in the powder, the flow property of HPMC increased due to the lubrication effect of surface moisture. The surface layer of HPMC was sheltered by a thick layer of moisture that enhanced its flow. Also, the flow properties of the hygroscopic drug, theophylline are affected to a greater extent due to water sorption [32].

#### 3.2.2. Compaction

Compaction refers to a process of achieving very intimate contact between two particles with an expectation to strongly bind them together and produce a sturdy particulate compact. The re-crystallization process essentially helps in achieving compacts of stronger strength by increasing the tensile strength and contact area of particles, thus reducing density variations within a tablet [52]. Absorbed water decreases the adhesion of tablets to the walls of the die due to its tendency to reduce the surface energy of particles. The moisture that emerges during compaction can play the role of a low-viscosity lubricant [53].

Low moisture-containing starches are not suitable for direct compression because of their poor compaction behavior [54]. Steendam et al. [44] reported the consequence of moisture on compaction behavior, porosity, and tablet strength due to a reduction in porosity and compactability, and an increase in tensile strength owing to particle bonding and decreasing elastic deformation.

#### 3.2.3. Tablet Hardness/Friability

Moisture uptake by tablets changes their hardness during storage. Uncoated tablets containing moisture-sensitive drugs are more prone to moisture [5]. Akbuga et al. [56] explained the effect of moisture on tablet hardness. The authors found that tablet hardness decreased two-fold at 75% RH compared to tablets stored at ambient RH. Nokhodchi et al. [55] examined the effect of moisture on ibuprofen tablets (Figure 7B). The compression force required during tableting was found to change for ibuprofen with varying concentrations of moisture content. The migration of bound water to the tablet surface during compaction reduced the interparticle bonding strength and increased elastic recovery. Viljoen et al. [57] reported the effect of humidity on the physical stability of chitosan powder and tablets. They reported discrepancies in the tablet hardness and friability after exposure to 40 °C/75% RH for six months.

#### 3.2.4. Dissolution

The dissolution of the drug is influenced to a greater extent due to the presence of moisture-sensitive drugs and excipients in the formulation. This may occur due to the reasons explained below [4].

##### Excipient Form Change

Some excipients used in the formulation are metastable and can convert into their stable form during storage due to exposure to high RH. The dissolution behavior of the dosage form containing such metastable excipients can be altered due to excipient crystallization [4]. Collier et al. [58] studied the effect of moisture and processing factors on the stability of levothyroxine sodium pentahydrate. Levothyroxine formulated with moisture-sensitive excipients such as sodium lauryl sulfate, povidone, and crospovidone resulted in its degradation. The mechanical properties of a disintegrant alter due to moisture uptake, which causes swelling of tablets leading to micro-cracks development. This affects the disintegrant effectiveness and finally its dissolution behavior. Bele and Derle, in their work, evaluated how the sorbed moisture affected the physical characteristics of disintegrants and their performance. In this study, various brands of extremely hygroscopic disintegrants such as Amberlite IRP 88, Doshion P 544 DS, Indion 294, and Tulsion 339 were used. They concluded that the moisture was responsible for plasticizing the disintegrants and reducing their yield pressures [59].

##### Gelatin Capsule Shell Changes

Hard gelatin capsule shells are unstable at high and low RH. At dry conditions, hard gelatin capsules become brittle at high RH, and the capsule shells become sticky and undergo chemical reactions that alter the dissolution behavior of drugs [60]. At conditions below 30% RH, the gelatin capsules become brittle and cracking issues are seen. Above 80% RH, the gelatin capsule shells become sticky and adhere to each other. Ofner et al. [61] explored cross-linking in gelatin capsules under stressed conditions. Soft gelatin capsules produced volatile agents at 37 °C and 81% RH that caused ɛ-amino group cross-linking, resulting in a reduced dissolution rate.

##### Polymorphic Transformations of API

Phase transition between solid phases is observed in hygroscopic drugs that affect the drug release process, dissolution. and flow property of powders (Figure 8).

**Hydrate/Anhydrate**: Water molecules may be present in APIs, either in a bound or unbound form. Generally, the water of hydration in crystals is reversible at a particular temperature and RH [4]. It is generally known that anhydrous forms and hydrates of the same API exhibit different dissolution profiles. Hence, any changes in the API’s hydration state during storage results in product quality failure [62].**Polymorph conversion**: Hydrate and anhydrous form interconversion is often reversible, but polymorphic form interconversions are usually irreversible. The primary concern for polymorph conversions is whether the polymorphic API form is the thermodynamically stable form during the whole storage temperature and humidity range [63]. Moisture can also induce polymorphic changes in some compounds, such as theophylline, and cause major batch inconsistencies [34].

**Amorphous to crystalline:** Amorphous and metastable forms generally have high volumes, high surface free energy, and weak intermolecular interaction. Hence, solvent association in such forms of solids is comparatively more than that of crystalline forms due to the easy breaking down of weaker intramolecular bonding by water. This is followed by the supersaturation of drugs, nucleation, and crystal growth of drugs. Hence, the amorphous form generally adsorbs significantly more moisture and converts into a crystalline form, leading to alterations in the dissolution profile [4,64].

## 4. Manufacturing Process-Related Issues of Moisture-Sensitive Drugs

Process-related issues have occurred during the manufacturing of formulations containing moisture-sensitive drugs and excipients. Manufacturing processes like grinding and drying can liberate bound water. Excipients with the ability to form hydrate can induce drug degradation as the water of hydration of excipients can take part in hydrolytic reactions during unit operations. Hence, during the manufacturing of hygroscopic drug formulations, any exposure to moisture leads to API degradation [65,66].

In the low-shear or high-shear wet granulation process, water is in direct contact with moisture-sensitive drugs and excipients and may cause the degradation of moisture-sensitive drugs. Hence, direct compression, roller compaction, or dry granulation methods are used to prepare granules of moisture-sensitive drugs. For low-dose drugs (<0.5 mg) minimum water exposure during fluidized bed granulation or wet granulation must be ensured or non-aqueous solvents can be used. Also, the granule/tablet coating operations can lead to hydrolytic reactions in moisture-sensitive APIs, especially during aqueous-based coatings. The moisture content increases the hardness of the tablets even at lower compression forces; this causes inconsistencies and requires extensive sampling, which increases the cost of manufacturing hygroscopic drugs [53].

Hygroscopic materials absorb moisture and increase moisture content in the final formulation, leading to microbial growth. It has been found that microbial growth decreased if the RH was <5%. At 80 to 100% RH, microbial growth rapidly increased. This is a serious issue in the ophthalmic and parenteral formulation, which further necessitates the inclusion of a preservative or additional unit operation such as free drying [67]. Moisture uptake during the manufacturing of tablets also causes sticking problems during the compression cycle [68]. Therefore, moisture has a significant effect on the granule characteristics, microbiologic stability, and subsequently the tablet properties during large-scale manufacturing. [69].

The moisture uptake by the co-crystals in tablets can induce the dissociation of the co-former from the API, leading to the loss of co-crystal characteristics. The presence of hygroscopic and ionizable excipients (such as magnesium stearate sodium starch glycollate, microcrystalline sodium, etc.) in a tablet can also induce co-crystal dissociation. This excipient-mediated co-crystal dissociation can have severe implications on the final product quality and leads to regulatory repercussions and batch failures. Hence it is advised to exclude such excipients that exhibit a tendency towards moisture uptake and cause dissociation of the labile co-crystals [70]. Veith et al. [71] developed a perturbed-chain statistical associating fluid theory (PC-SAFT) to predict the stability of co-crystals at higher humidity conditions. From the same model, they found that the presence of sugar excipients such as fructose and xylitol can greatly reduce the deliquescence RH of co-crystals, thereby reducing the stability of the co-crystals.

## 5. Co-Crystals Approach to Improve Hygroscopic Stability

Co-crystals can reduce hygroscopicity and also improve the stability of APIs while maintaining their aqueous solubility profile. This is a major advantage of co-crystals. Various examples are available wherein the stability enhancement occurs without any serious solubility concerns in the molecules [72,73,74]. There are numerous instances where better moisture stability has been attained through the co-crystallization process [16,36,75,76]. As discussed earlier, the free hydrogen bonding sites are mainly responsible for the moisture uptake. In co-crystals, the hydrogen bonding sites are occupied and are thus largely responsible for the improved resistance of the molecule for moisture uptake [75]. Liu et al. [77] showed that the use of neutral co-formers is advantageous when the objective is to reduce the hygroscopicity of the molecules. The neutral co-formers exhibit self-assembly properties during crystal structure formation that occupy the hydrogen bonding sites as well as shield the API from bonding to the water molecules [23]. There are many studies where researchers have tried to identify the underlying reason behind this enhanced stability. Other fine structural aspects or changes that improve the stability of co-crystals against hydration are outlined below.

### 5.1. Molecular Orientation and Aromatic Interactions

In this section, we attempt to explain the probable mechanisms by which the co-crystals can have enhanced stability against hydration/moisture uptake. The orientation of the molecules and the geometry of the arrangement in a molecule greatly influence the physicochemical properties of a crystal form. The differences in the molecular arrangement arising from different orientations result in the charge redistribution across the molecule and change the lattice energy, which in turn alters the bonding interactions of the molecule (e.g., hydrogen bonding). The aromatic donor-acceptor interactions also influence the atomic charge on the atoms of the molecule that are not involved in any bonding interactions [78]. Stanton et al. [28] carried out a charge density distribution study to understand the atomic interactions responsible for reducing the hygroscopicity of the triclinic malonic acid-theophylline co-crystals. They studied various parameters, such as the orientation of the co-former and API molecules, lattice energies, and molecular interactions in different crystalline forms of the same co-crystal, i.e., monoclinic and triclinic forms. The ends of the co-former and API molecules facing each other were found to have the most favorable orientations, which enabled the aromatic interactions between two API molecules and thereby contributed to the stabilization of the molecule as found in the triclinic form. At the same time, no such molecular configurations were seen in the monoclinic form of theophylline-malonic acid co-crystal. Figure 9 depicts the difference in the arrangement of co-crystal molecules of triclinic and monoclinic forms. The malonic acid acts as both a hydrogen bond donor and acceptor and as a bridge between the theophylline molecules. The packing density of a crystal is mainly dependent on the strength of the intermolecular bonding forces present between the molecules [79]. The inefficient molecular interactions can lead to defects in the packing of a crystal and thereby gives rise to large void spaces in the crystal lattice of a compound. The void spaces in the crystal lattice provide a favorable condition for hydration under different storage conditions [80]. The stronger hydrogen bonding of malonic acid and theophylline in this way was the net result of charge density redistribution. The influence of strong hydrogen bonds on reduced hygroscopicity was also noticed by Qi et al. [36] in their study. Also, the aforesaid orientation of the molecules makes it possible for the aromatic rings to interact with each other through π stacking and aromatic interactions that favorably increase the molecular packing density and reduce the void space volume. The π stacking is known to reduce the hygroscopic character of a compound [77]. Thus, the greater the π interactions in a molecule, the higher the stability of the molecule against hydration.

Hirshfield analysis of the above co-crystal forms indicated that the electronegative -N and -O atoms were occupied (due to bonding) to a greater extent in the triclinic form than in the monoclinic form, which represents the presence of stronger hydrogen bonding in the triclinic form. The bridging of the malonic acid between the API molecules provides high stabilization energy and covers up the void space in the crystal form (Figure 10). These aspects give the triclinic co-crystal form greater stability against moisture uptake compared to the anhydrous theophylline and monoclinic co-crystal. Also, in the theophylline anhydrate structure, only two hydrogen bonding sites are occupied but in the case of the malonic acid-theophylline co-crystal, up to eight hydrogen bonding sites are utilized in the structure formation. This feature also indicates the participation of the free hydrogen bond donor/acceptor present in the molecule in the moisture uptake by the molecule [28].

### 5.2. Correlation with Lattice Energy and Its Measurement

Lattice energy refers to the energy that is required to break a crystalline structure into its individual components. Lattice energy comparisons have been previously reported for predicting the formation of hydrates of a particular compound [81]. This parameter can indirectly approximate the hygroscopic stability of a molecule or co-crystal system. The CE-B3LYP model energies can be calculated using the tools available in CrystalExplorer software. The co-crystal having lower lattice energy is known to have reduced hygroscopicity when compared to the co-crystal with higher lattice energy [82]. Lattice energy calculations can also be used to predict the stability of the co-crystals. The lattice energy of triclinic and monoclinic forms of theophylline-malonic acid co-crystals are −231.7 kJ/mol and −267.6 kJ/mol, respectively. This infers that the monoclinic form is more stable compared to the triclinic form of theophylline-malonic acid co-crystal, which can be regarded as the metastable form [28,83].

Stanton et al. calculated the lattice energy of the co-crystal using CrystalExplorer software by measuring the energy between the central and different molecules within a defined radius of 25A° using the B3LYP function and 6-31G (d,p) basis set. The software calculates the coulombic, repulsion, dispersion, and polarization energies of a molecule and the sum of these forces provide the lattice energy for a given molecule. The works of Thomas et al. [84] and Mackenzie et al. [85] have provided a method for calculating the lattice energy and intermolecular interaction energies, respectively, using the CE-B3LYP level of theory [28].

### 5.3. Hygroscopic API and Hydrophobic Cofomer

A plethora of co-formers are available, which can be selected based on their properties for incorporating desired characteristics into the co-crystals. The previous subsection explained the interactions between the API and co-former through charge density studies that can also be used as guiding tools for the selection of suitable co-formers to improve a particular property [28]. The use of hydrophobic co-formers for the co-crystallization of a hygroscopic API seems like a direct approach to improve API stability and has been reported numerous times in the literature [40,75,86]. It has been mentioned that a more hydrophobic co-former and a stronger crystal lattice might also contribute to the reduced hygroscopicity of co-crystals. Nicotinamide (NCT) is a highly hygroscopic substance and shows a high affinity for moisture due to the presence of free nitrogen (hydrogen bond acceptor) in its structure. Co-crystallization of NCT with the hydrophobic co-former, IBU, decreases the moisture sorption tendency multi-fold. It has been observed that in the co-crystal of ibuprofen (IBU) and NCT, the free pyridine nitrogen atoms of NCT form weak hydrogen bonds with the carboxylic hydrogen atoms of IBU. This interaction renders the free nitrogen atoms of NCT unavailable for bonding with water molecules, decreasing their tendency for moisture sorption. Similarly, BCl [40] and theophylline [75] have been co-crystallized with emodin and apigenin, respectively, which greatly improved their stability against moisture uptake. L-carnitine undergoes deliquescence above 60% RH, making it highly difficult to formulate as a tablet. Pang et al. [87] prepared co-crystals of L-carnitine with the hydrophobic co-former myricetin to improve processability. The prepared co-crystal of L-carnitine and myricetin significantly reduced the moisture uptake (0.27%) at 80% RH. The strict hindrance by hydrophobic myricetin is believed to be involved in reducing the moisture uptake by L-carnitine. These works show the importance of hydrophobic co-former in controlling the hygroscopicity of moisture-sensitive drugs. However, care must be taken that the hydrophobic co-former does not interfere with the solubility of the co-crystal [36,40].

### 5.4. Recent Works

While preparing the co-crystals of an API, the choice of the co-former plays a vital role. Not all the co-formers can improve the solubility or stability of an API. The interactions between the co-former and the API decide the final characteristics of the co-crystal. For instance, Stanton et al. [73] examined the physicochemical properties of the co-crystals of non-hygroscopic drug candidate AMG517. Among the 10 co-crystal forms studied, the AMG517-2-hydroxy caproic acid co-crystals showed a slightly higher hygroscopic propensity, while the others were non-hygroscopic. The co-crystals of ciprofloxacin with nicotinic acid and isonicotinic acid also showed hygroscopicity as assessed by the FTIR spectra of the co-crystals [88]. This shows similarity to the case of nicorandil and its co-crystals of salicylic acid, 1-hydroxy,2-naphthoic acid, 3-hydroxybenzoic acid, and 2,5-dihydroxybenzoic acid, respectively. Herein the co-crystals showed a slightly increased hygroscopicity but had better storage stability in terms of API degradation compared to the pure nicorandil API [89]. Hence, it is not to be misunderstood that the co-crystal formation always reduces the hygroscopicity of a compound. The co-former selection plays an important role in achieving higher hygroscopic stability. A research group attempted to study the hygroscopicity of organic molecules and found that this depended on the functionalities present in the molecule. The polarity of the functional groups affected the hygroscopicity of the compound in the sequence -CH_3_ < -NH_2_ < -OH < -CHO < -NH_2_OH < -COOH. The position of the functional groups also influenced the hygroscopicity, e.g., fructose has hydroxyl groups away from the C=O group and was more hygroscopic than mannose which contained these two groups closer to each other. The hygroscopicity of the compounds increases with an increase in the O/C ratio in the organic compounds [90]. There have also been comparative studies of co-formers in the same fashion, which facilitate their selection to mitigate the hygroscopicity issues of API. Numerous works have been presented here whose primary focus was to reduce the hygroscopicity of APIs such as oxymatrine, theophylline, indomethacin, caffeine, etc [36,75,91,92,93].

Likewise, caffeine/oxalic acid co-crystals have been demonstrated to be superior to caffeine anhydrate in terms of their physical stability to humidity. In this setting, the hydrogen-bonding sites that participate in the formation of caffeine monohydrate are used to connect the co-former (oxalic acid) with the API molecule, thus blocking the possibility of caffeine hydrate formation during processing and storage [22]. Co-crystals of isosorbide (ISO) with the co-formers piperazine (ISO-PZ), hydrochlorothiazide (ISO-HCT), 3,5-dihydroxybenzoic acid (ISO-DHBA), and gallic acid (ISO-GA) were prepared by Watanabe et al. [8]. They observed that all the co-crystals were less hygroscopic when compared to ISO crystals. Co-crystal structure investigations revealed that the amount of water absorbed by the crystals was directly related to the hydroxyl groups (hydrogen bond donor/acceptor) exposure on the crystal surface.

Co-crystallization of caffeine with various carboxylic acids has been performed and the results demonstrated that the obtained co-crystals possess less moisture uptake tendency than API. The samples were subjected to four RH conditions and collected and analyzed after 1, 3, and 7 weeks. Of all the co-crystals with caffeine, the best moisture stability was elicited by the caffeine-oxalic acid co-crystal under all RH conditions [22]. The deliquescent oxymatrine (OMT) was co-crystallized with urea, sulfanilamide (SUA), theophylline (THP), 2-keto glutaric acid (KTA), and 3-Hydroxy-2-naphthoic acid (HNA) in order to improve its stability against moisture uptake (Table 4). Among the five co-formers used, urea further increased the hygroscopicity of the compound while the SUA, KTA, and HNA co-crystals showed improved stability against moisture uptake. Of all the OMT co-crystals, the one with KTA was most desirable due to its good solubility, intrinsic dissolution rate (IDR), and improved hygroscopic stability at 95% RH [36]. Table 5 presents examples of various hygroscopic substances for which the co-crystallization technique was used to improve their physical stability.

Co-crystals providing stability to moisture-sensitive molecules are not only limited to pharmaceuticals but also other chemical-based industries, for instance, pesticides. Chen et al. [94] observed that the humidity stability of pymetrozine (PMZ) was improved by the co-crystals. It was observed that the hydrogen bond strengths in co-crystals were significantly higher than those in pure PMZ and PMZ·2H_2_O, which was the reason behind the improved humidity stability of the co-crystals. PMZ co-crystals showed improved stability against moisture because of the strong hydrogen-bonding networks. Sun et al. [18] studied the applicability of CAB (conjugated acid-base) co-crystals over corresponding benzoic acid and its salts. They observed that CAB co-crystals stopped the degradation of ammonium benzoate and inhibited the deliquescence of sodium benzoate and potassium benzoate.

Temozolomide is a moisture-sensitive drug with a tetrazine ring that undergoes hydrolytic cleavage in the presence of water, which has been shown in the following studies. The susceptible group of hydrolytic cleavage was not exposed after the formation of co-crystal with oxalic acid and salicylic acid [95]. Karangutkar et al. utilized the co-crystal approach to stabilize the betacyanins obtained from the fruit extract of *Basella rubra* and compared this with freeze-dried extract; the freeze-dried extract was a sticky mass and was highly hygroscopic. Conversely, the sucrose and gum acacia co-crystallized extract of *Basella rubra* had good flow properties, reduced hygroscopicity, and showed good storage stability [27]. In an attempt to improve solubility and reduce the hygroscopicity of phenazopyridine hydrochloride, Tao et al. co-crystallized phenazopyridine with phthalimide (Figure 11). The hydrochloride salt of phenazopyridine showed ~27% weight gain at 98% RH, while the co-crystal form was very stable with only ~5% weight gain at 98% RH after 28 days of exposure. The co-crystal shows packing of scissor-like chains [96].

**Table 5 pharmaceutics-15-00189-t005:** List of co-crystals reported to prevent moisture-sensitive APIs.

Drug	Coformer	Synthon	Observation
Indomethacin	Saccharin	The acid and imide dimers of API and co-former are interconnected by an N–H-O hydrogen bond	Moisture uptake by co-crystal was almost negligible (<0.05%) at 98% RH than the stable indomethacin [97]
Theophylline	Flufenamic acid	API and co-former connected by an acid-imide heterosynthon involving O−H···O and N−H···O hydrogen bonding. The C−H···F interactions lead to the generation of ladder networks.	Hygroscopicity of theophylline was decreased by co-crystal formation [91]
* S * -oxiracetam	Gallic acid (GA)	Tetramer formation is caused by the intermolecular hydrogen bonding between O−H···O (at 4 places) and N−H···O (at 2 places). Construction of R^2^_2_(9) synthon is by amide and two hydroxy groups of phenol.	Significant improvement in the hygroscopic stability of S-oxiracetam-GA co-crystal at 98 % RH [98]
Caffeine	Oxalic acid	Caffeine and oxlic acid connected through a heteromeric synthon of O-H-N C-H-O hydrogen bonding	The co-crystal of caffeine with oxalic acid was completely stable at 98% RH for several weeks compared to pure caffeine [92]
Temozolomide (TMZ)	Oxalic acid Salicylic acid	The carboxamide group of TMZ and carboxylic acid of the co-former is connected through hydrogen bonding via amide–acid hetero synthon	Both the co-crystals of TMZ show longer half-lives (2 times greater) compared to the TMZ alone [95]
Flucytosine (FLC)	Gallic acid (GA) Glutaric acid (GLA) 2,3-dihydroxy benzoic acid (2,3 HBA)	API and co-formers are interconnected through hydrogen bonding between O-H∙∙∙O, N-H∙∙∙O, and N-H∙∙∙N homosynthons	FLC-2,3HBA, FLCGAA, and FLC-GLA exhibited higher -% RH and stability at both 70–75% RH and 90–95% RH conditions [99]
Tramadol	Paracetamol	Interconnected through hydrogen bond between Cl of tramadol to H of paracetamol	Hygroscopicity of co-crystal was decreased [100,101]
Oxymatrine (OMT)	Urea Sulfanilamide (SUA) 2-Ketoglutaric acid (KTA) 3-Hydroxy-2-naphthoic acid (HNA) Theophylline (THP)	The API and co-formers are interconnected through either of O-H∙∙∙O, C-O∙∙∙H, N-H∙∙∙O^−^-N^+^, N-H∙∙∙O=C hydrogen bonding. In the case of urea co-crystals, the API and co-former are interconnected through the water molecule in between.	Hygroscopicity of the co-crystals was in the following order: Urea > THP > KTA > HNA > SUA [36]
Phenazopyridine hydrochloride	Phthalimide	Drug and co-former forms a dimer with O-H∙∙∙N, N-H∙∙∙N hydrogen bonds. Furthermore, the dimers are linked together, forming scissor-like chains.	Hygroscopic stability of phenazopyridine greatly improved after co-crystal formation [96]

Chow et al. [86] prepared nicotinamide co-crystals with ibuprofen and flurbiprofen, which reduced the hygroscopic property of nicotinamide. Nicotinamide has two moisture-susceptible groups, i.e., amide and pyridine ring with nitrogen; both groups are free to react with water by hydrogen bonding, thus contributing to their hygroscopic nature. Ibuprofen and flurbiprofen form co-crystal with nicotinamide, and react with pyridine nitrogen of nicotinamide by forming a hydrogen bond with them. Therefore, not all the groups involved in hydrogen bonding with water can reduce the moisture sorption at relatively low RHs.

## 6. Methods and Tools for Moisture Content Determination

The ill effects of moisture on the stability of pharmaceuticals can be understood perfectly only when the exact amount of moisture in a substance can be quantified by a validated method. Its estimation can then be used as a tool to correlate the extent of the deleterious effects of moisture on a finished product. Perhaps loss on drying is the most common and simple method used to approximate the water content in samples [102]. The Karl–Fischer (KF) titration method gives reliable water content in liquid samples and is commonly used for water content determination at the laboratory scale. There are several modifications reported in the literature to improve the efficiency of the KF method to determine moisture content [103]. Similarly, various other methods have been developed and used to determine the moisture content in the samples [104,105]. But these methods usually require some sample preparation, and thus are time-consuming. The advent of the NIR and related tools to measure water content immediately in samples during the process can provide a lot more information about the sample’s hygroscopicity compared to conventional analytical methods. This is facilitated by Process Analytical Technology (PAT). According to the Food and Drug Administration, “PAT is a system for designing, analyzing, and controlling manufacturing through timely measurements (i.e., during processing) of critical quality and performance attributes of raw and in-process materials and processes with the goal of ensuring final product quality.” Essentially, PAT provides an enhanced understanding of the unit processes and thereby facilitates process and quality improvement [106,107]. Various method and tools available for moisture content determination are listed in Table 6.

Non-invasive techniques such as near-infrared (NIR) [108] and microwave resonance (MR) [109] sensors are utilized for the well-timed monitoring of critical quality attributes (CQA) of materials and the execution of PAT. The calibration is time-consuming and chemometrics can be used. In addition, the use of calibration sets developed with minimum standards can be beneficial. Monitoring minute amounts of residual moisture can be possible by using NIR or MR to yield superior process control and endpoint determination. The implementation of PAT tools can assist in the timely release of the batch. These probes can be fitted to any reactor and online monitoring of moisture/water content in samples is possible. Jorgensen et al. [108] employed the NIR and charge-coupled device (CCD) Raman spectroscopic methods to monitor the hydrate formation during the wet granulation of theophylline and caffeine. They concluded that Raman spectroscopy allows for the monitoring of the pseudopolymorphs through changes seen in the drug molecule itself. The NIR spectroscopy was superior to Raman as it could differentiate between the free and hydrate water.

**Table 6 pharmaceutics-15-00189-t006:** Principle advantages and limitations of various methods for determining the moisture content in samples.

Method	Principle	Advantages	Limitations
Loss on drying [110]	Works on the thermogravimetry principle. Material is heated until no more weight is lost and the final loss of weight is calculated.	Simple and convenient Standard Conventional method Rapid	Measures both water and volatile impurities i.e., not water-specific Decomposition of sample Results are dependent on drying time, exposure to the environment, and weighing accuracy
Karl Fischer titration [103,110]	Water reacts with iodine and reaches the endpoint when all the water is consumed	Highly specific and selective to water content in a product sample Determinations ranging from percentage to ppm levels	Sample destruction Poorly soluble and reactive samples cannot be analyzed
Near Infrared spectroscopy [111]	Absorption of electromagnetic radiation by the sample, and its transmittance and absorbance are measured by the detector	Rapid, precise, non-contact and non-destructive In-process quality control Minimal sample preparation	Needs initial calibration Measurement of moisture from the surface of a sample
Raman spectroscopy [108]	The Raman shift is unique to each molecule and water being a weak Raman scatterer can be identified and quantified. It is also possible to quantify the water content from the Raman intensity.	Simple, rapid, and nondestructive Analysis can be carried out in different physical states: liquid, pastes, solids, etc.	Requirement of customized feeders and sample collectors High laser power is not suitable for easily dehydrating materials
Microwave technology [109]	When microwave interacts with water within the material they slow down and weaken attenuate as the energy is transferred to the water. The signal is received by the receiver antenna and is compared to the transmitted signals.	Non-contact and non-destructive Rapid and accurate Moisture content measurement is representative of the entire product and is not just a measurement of surface moisture In-process quality control	Possible leakage of microwave energy during the measurement Needs initial calibration Results can be affected by particle size, temperature, polarization
Dynamic vapor sorption [112]	The gravimetric technique measures the amount of solvent absorbed with respect to its mass by varying vapor concentration surrounding the sample	Precise, robust, and gives reproducible results High sensitivity Solid-state changes in the sample can be identified Requires less sample and time	Instrument cost is high
Dielectric capacitance [110]	Measures the difference between the dielectric constant of water and the material located between the sensor’s capacitor electrodes	Least expensive Rapid and non-destructive Accurate Highly sensitive due to the large dielectric constant of water	Difficulty to measure bound water at high frequencies Calibration is required for individual product sample

## 7. Future Perspectives

The pharmaceutical industry and regulatory bodies have openly accepted the co-crystal approach to improve the stability of drugs, which is evident from the approval of Steglatro^®^ [ertugliflozin and L-pyroglutamic acid (1:1)] in 2017 [110]. Ertugliflozin is a highly hygroscopic drug and its co-crystal formation results in the reduction of its hygroscopicity. It is expected that more co-crystal formulations exhibiting higher stability will be in the marker in the future. To achieve this, the selection of co-former plays a crucial role. It is necessary to correctly identify the co-former that can favorably interact with a particular API and improve its physicochemical properties and its stability profile. Few research works have shown that the presence of aromatic interactions and stronger hydrogen bonding interactions between API and the co-former is beneficial for reducing hygroscopicity [77,79]. There can be model-based predictions to guide the selection of suitable co-formers based on the possible interactions for a given molecule. This shall eliminate the trial and error-based methods of co-former selection and revolutionize the co-crystallization approach.

## 8. Conclusions

For an effective drug formulation, the study of API-moisture interaction is crucial along with the API-excipients interactions, and holds true, particularly for drugs that are hygroscopic and moisture-sensitive, which may account for half of those currently on the market.

Moisture can enter in dosage forms from numerous sources such as bulk drugs, excipients, manufacturing processes, and environmental conditions. Moisture can interact with drugs in different ways, including adsorption on the surface, capillary condensation, formation of crystal hydrate, or undergoing complete deliquescence. This has direct and indirect effects on the stability of pharmaceutical products. Hence consideration of hygroscopicity plays an important role in the formulation development of such materials and analyzing the materials for the moisture content is essential during new formulation development to predict instability issues and problems that occur due to the hygroscopicity of materials. Moreover, this may have significant effects on tablet compaction, wet granulation, powder flow properties, and microbial growth. Based on the literature survey, basic ideas on classification, determination of moisture content, the impact of moisture content on the hygroscopic drugs and excipients and co-crystal strategies to beat the effect of its mechanism, have been discussed.

In a pharmaceutical system, once issues due to the hygroscopic nature of materials are understood, the necessary action can be taken to address the unwanted reactivity, and thus increase the stability and improve the efficiency of the final products. Identifying how the physicochemical properties of materials allow them to interact with moisture is needed to better understand them. Modern approaches to characterizing hygroscopic compounds that go beyond the routine moisture sorption–desorption measurement and total knowledge of water–solid interactions at a molecular-level shall help build robust strategies for handling and processing pharmaceutical solids. Furthermore, the use of the PAT tool for determining moisture content needs to be applied for the process control of moisture-sensitive drugs. It is possible to reduce the probability of instability in pharmaceutical products due to hygroscopicity by applying knowledge of mechanisms and methods for solving it, as we have tried to explain in our discussion of the changes in the molecular structures in co-crystals. We hope that this review provides some perspective on the significance of hygroscopicity and the mechanism by which co-crystals formation can mitigate the hygroscopicity issues of pharmaceutical APIs.

## Figures and Tables

**Figure 1 pharmaceutics-15-00189-f001:**
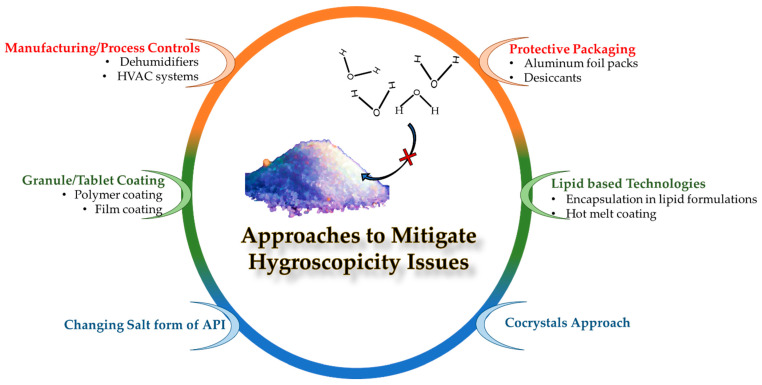
Illustration of methods to improve the hygroscopic stability of moisture-sensitive APIs.

**Figure 2 pharmaceutics-15-00189-f002:**
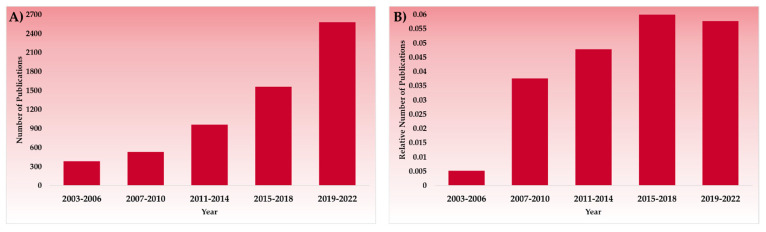
Trends of scientific publications from 2003 to 2022 for (**A**) Drug stability, hygroscopicity, (**B**) Co-crystals, drug stability, hygroscopicity. Data taken from Science Direct.

**Figure 3 pharmaceutics-15-00189-f003:**
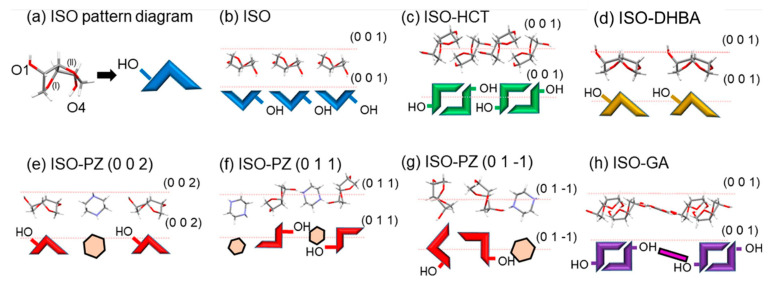
Schematics of the major surface structures of the ISO crystals and co-crystals. Pink dotted lines representing the major crystal faces are shown to indicate the faces that were horizontal and perpendicular to the paper sheet. The crystal face structures are shown with the stick models. Reprinted with permission from [8].

**Figure 4 pharmaceutics-15-00189-f004:**
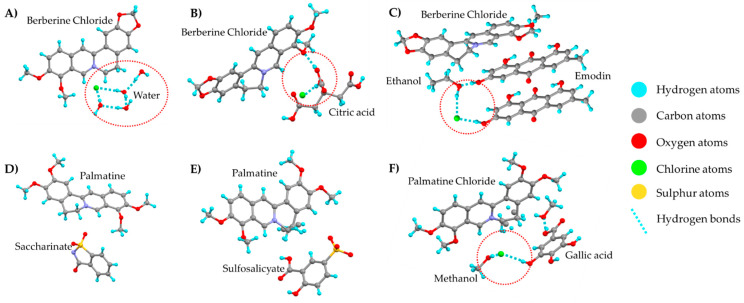
(**A**) Berberine chloride tetrahydrate (#1306671), (**B**) Berberine chloride-citric acid co-crystal (#1857453), (**C**) Berberine chloride-2emodin-ethanol co-crystal (#1862517), (**D**) Palmatine saccharinate salt (#1977091)), (**E**) Palmatine sulfosalicyate salt (#2053643)), (**F**) Palmatine-gallic acid co-crystal (#2075058) (from Mercury 2022.2.0, Build 353591). #—CCDC Identifier number.

**Figure 5 pharmaceutics-15-00189-f005:**
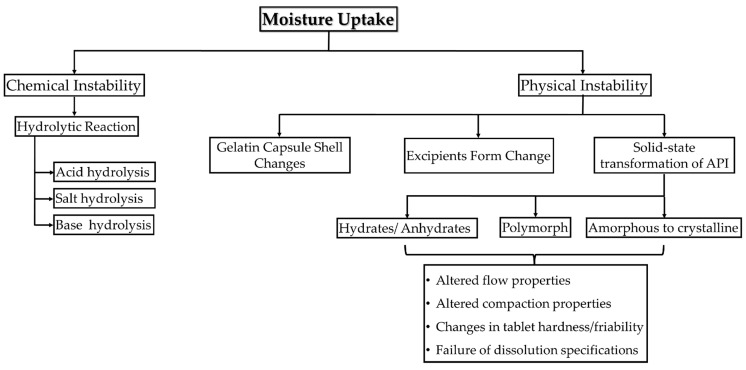
Schematic chart showing the chemical and physical instabilities caused due to moisture ingress in an API.

**Figure 6 pharmaceutics-15-00189-f006:**
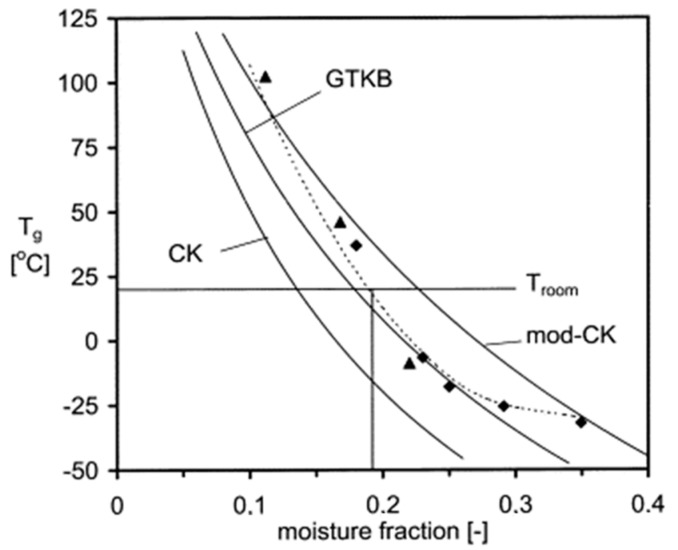
Variation of T_g_ of amylodextrin with moisture fraction. Experimental data representing the glass transition as determined from conventional DSC (♦) and the reversing heat flow of modulated DSC (▴). Reprinted with permission from [44].

**Figure 7 pharmaceutics-15-00189-f007:**
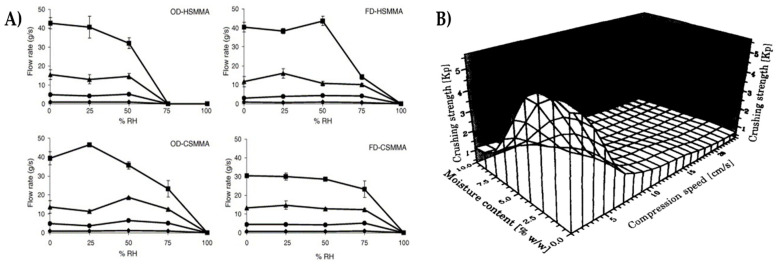
(**A**) Flow rate (g/s) vs. relative humidity percentage (RH) of the different methacrylate copolymers using stainless-steel funnels of 20 mm (■), 15 mm (▴), 10 mm (•). and 5 mm (♦) holes. Error bars represent the standard deviation, (**B**) Correlation between crushing strength, compression speed, and moisture content for 400 mg ibuprofen tablets. Reprinted with permission from [49] and [55], respectively.

**Figure 8 pharmaceutics-15-00189-f008:**
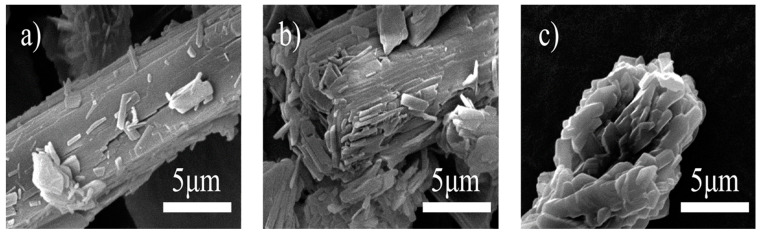
SEM pictures of theophylline: (**a**) Form III crystals (before transition), (**b**) Half way during transition, (**c**) Form II crystals (after transition). Reprinted (adapted) with permission from [62]. Copyright 2007, American Chemical Society.

**Figure 9 pharmaceutics-15-00189-f009:**
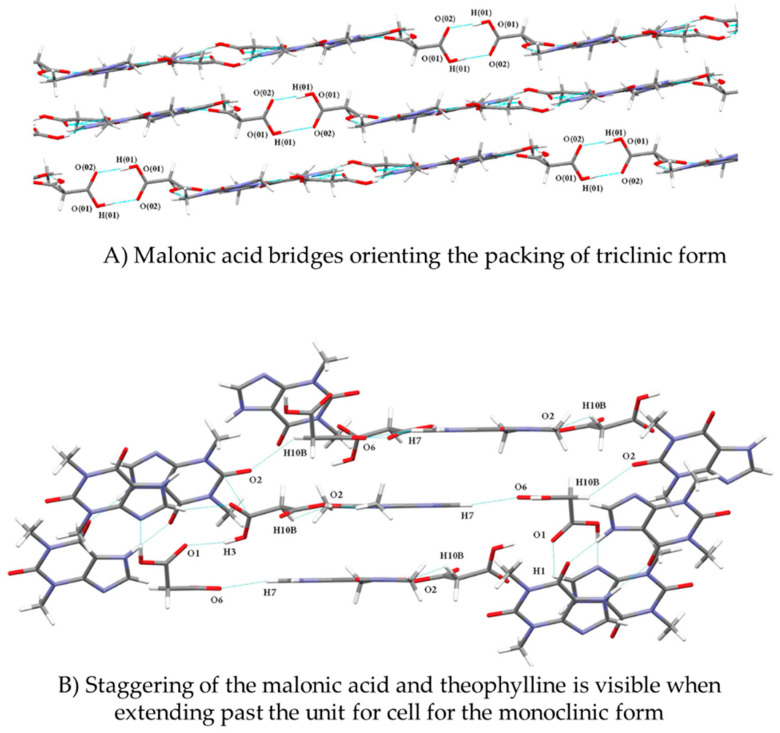
Theophylline-malonic acid co-crystals of (**A**) Triclinic, and (**B**) Monoclinic form, showing the difference in the malonic acid orientation. Reprinted (adapted) with permission from [28]. Copyright 2021, American Chemical Society.

**Figure 10 pharmaceutics-15-00189-f010:**
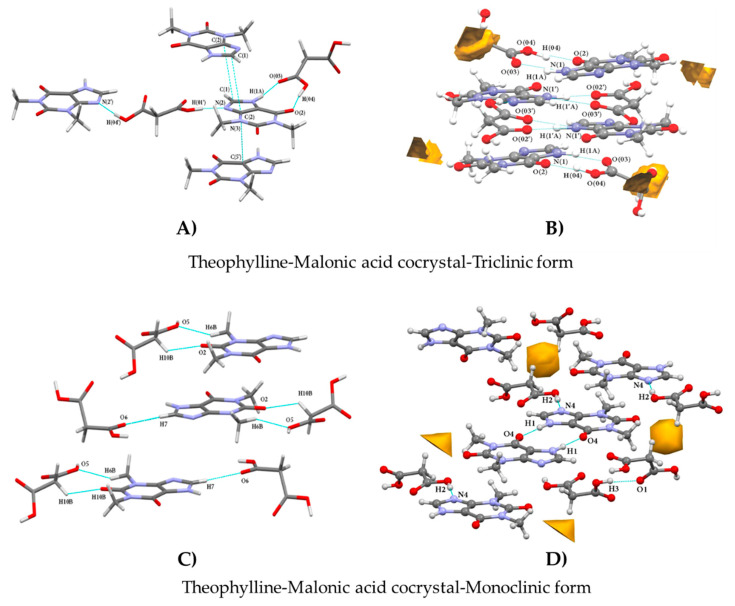
(**A**,**C**)—Hydrogen bonding and aromatic stacking, (**B**,**D**)—Void space present within the co-crystals. Reprinted (adapted) with permission from [28]. Copyright 2021, American Chemical Society.

**Figure 11 pharmaceutics-15-00189-f011:**
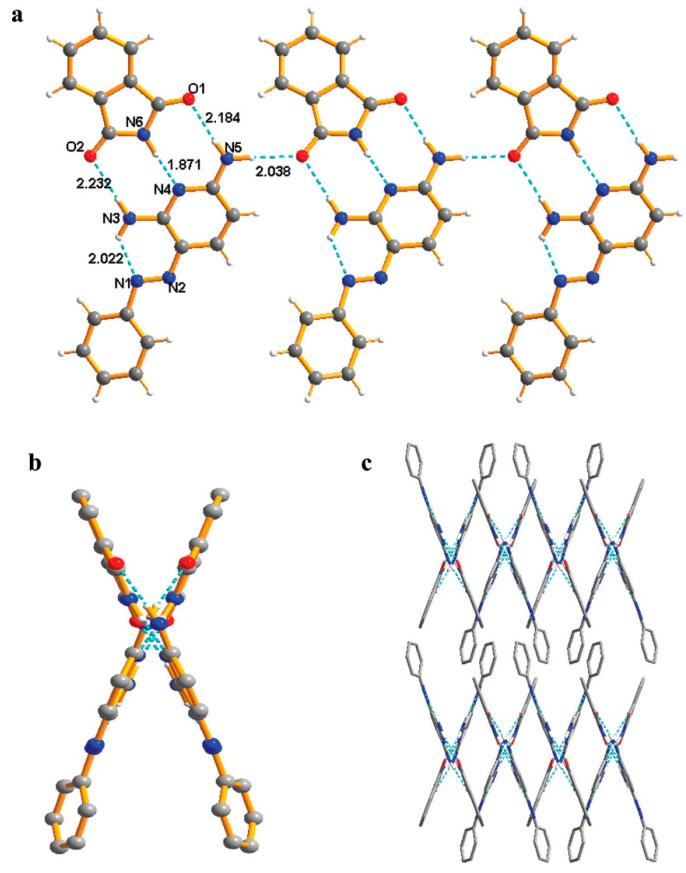
Phenazopyridine-phthalimide co-crystal structure. (**a**) Side view of hydrogen bond linked chain, (**b**) Top view of scissor-like chain, and (**c**) 3D structure of co-crystal. Reprinted (adapted) with permission from [96]. Copyright 2012, American Chemical Society.

**Table 1 pharmaceutics-15-00189-t001:** Advantages and disadvantages of various approaches to control the hygroscopicity issues of API.

Control Strategy	Advantages	Disadvantages	References
Manufacturing Controls (dehumidifiers and HVAC)	Humidity control can be achieved over a large area with good accuracy With artificial intelligence, the humidity can be automatically controlled and customized programming is possible based on the requirement	Requires time to achieve desired relative humidity. Difficult to achieve <25% RH Not an economical approach	[11,20]
Protective Packaging (aluminum foil pack and use of desiccants)	Non-toxic and non-reactive with the product Stable over a longer temperature range Provides a complete barrier to any vapor and gases	Pin holing is a common disadvantage When the strip pack is stretched or folded, aluminum can undergo splitting (flex cracking) Aluminum extraction and use of desiccants is a costly affair	[12]
Polymer/Film Coating	Provides a good degree of protection against moisture ingress Can also provide sustained and pH-responsive release profiles	Coating is an additional unit operation A skill extensive operation that requires specialized equipment Requires usage of organic solvents	[21]
Lipid-based Technologies	Provides controlled release of the drug Increases the absorption of poorly soluble drugs	Requires specialized storage conditions Increases the cost of the final product Lipid degradation products are toxic Formulation development requires expertise and specialized equipment	[14]
Changing Salt forms of API	Greatly improves the aqueous stability of drugs Can be carried out during API synthesis Can also modify other physicochemical properties of the API	Risk of reduction in the aqueous solubility of drugs Requires the presence of an ionizable group Limited number of counter ions for a given compound	[15,19]
Co-crystals	Not only reduces the hygroscopicity but also improves the overall stability of the molecule No requirement of ionizable groups Variety of co-formers can be used, and they are non-toxic Co-crystals of almost all types of APIs can be prepared	Requires the presence of hydrogen bond donors/acceptors to form co-crystals In some cases, a reduction in solubility is noted	[22,23]

**Table 2 pharmaceutics-15-00189-t002:** Water vapor adsorption amount per unit surface area (mg/m^2^) of ISO and co-crystals at 30% RH. Reprinted with permission from [8].

Substance	Critical Relative Humidity (%RH)	Water Adsorbed at 30% RH/Unit Surface Area (mg/m^2^)
ISO	48	1.636
ISO-PZ	56	0.572
ISO-HCT	85	4.963
ISO-DHBA	69	10.113
ISO-GA	67	1.172

**Table 3 pharmaceutics-15-00189-t003:** Comparison of hygroscopic data of berberine chloride, palmatine chloride, and adiphenine hydrochloride with their co-crystals or different salt forms.

Compound	Coformer	% Weight Gain (*w/w*)		
		70% RH	95% RH	Changes in Solubility
Berberine chloride (BCl) [16,17]	--	8.80%	16.30%	--
Citric acid-berberine	Citric acid [16]	< 2.0%	8.00%	Co-crystal had similar solubility as that of BCl
Emodin-berberine chloride (EM-BCl)	Emodin [40]	~ 0.50%	1.30%	Solubility in the order: BCl > EM-BCl > 2EM-BCl-Et
2 Emodin-berberine chloride-ethanol (2EM-BCl-Et)		~ 0.75%	0.90%	
L(+)-Lactic acid-berberine chloride	L(+)-Lactic acid [17]	~ 4.00%	10.00%	Co-crystal had similar solubility as that of BCl
Palmatine chloride [35]	--	8.14%	19.00%	--
Palmatine-saccharinate (Salt) [35]	--	3.72%	2.30%	Solubility of new salt reduced
Palmatine-sulfosalicyate (Salt) [19]	--	5.90%	13.43% ^#^	Solubility of new salt reduced
Gallic acid-palmatine chloride [23]	Gallic acid	2.83%	5.78% ^#^	Solubility of co-crystal reduced
Adiphenine hydrochloride [15]	--	~ 2.5%	22% ^@^	Retained 5% water during desorption even at 0% RH
Adiphenine citrate (Salt) [15]	--	~ 0.75%	3.2% ^@^	Significant reduction in aqueous solubility
Adiphenine oxalate [15]	--	~ 0.25%	2.6% ^@^	Significant reduction in aqueous solubility

# at 98% RH conditions; ^@^ at 90% RH conditions.

**Table 4 pharmaceutics-15-00189-t004:** Co-crystals of oxymatrine showing the IDR, solubility, and hygroscopicity data (reprinted with permission from [36]).

Co-Crystal	IDR (mg cm^−2^ min^−1^)	Solubility (mg ml^−1^)	Hygroscopicity
OMT-Urea-2 H_2_O	30.1	>48	87.69%
OMT-SUA	11.4	>52	0.97%
OMT-THP	2.8	>54	45–50%
OMT-KTA-H_2_O	21.7	>50	6.06%
OMT-HNA	0.1	2.92	2.45%

## Data Availability

The data presented in this study are available in the article and Appendix A.

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
