# Peer review of "Co-Crystallization Approach to Enhance the Stability of Moisture-Sensitive Drugs"

_pharmaceutics, 2023, doi:10.3390/pharmaceutics15010189_

Round 1

Reviewer 1 Report

Please double-check the manuscript  and correct typos and mistakes like

“Europian journal of pharmaceutical science”

   “CE-B3LYP program of the CrystalExplorer software”  - CE-B3LYP is not a program, but a level of theory

 “using the B3LYP function and 6-34G (d,p) basis set”   -  not 6-34G (d,p) but 6-31G (d,p)

Both “Co-crystal” and  “cocrystal” is used in main text. Please decide in favour of one of them.

 There are some more relevant works that can be mentioned (doi 10.1039/D0CE01820A, etc)

Author Response

The authors would like to thank the reviewer for the suggestions and for helping improve the manuscript's quality.

  1. “Europian journal of pharmaceutical science.”

Response: The typo in reference no. XX has been corrected to “European” as per the suggestion (reference no. 44 of the updated manuscript.

  1. “CE-B3LYP program of the CrystalExplorer software”- CE-B3LYP is not a program, but a level of theory.

Response: Replaced “program” with “level of theory”.

  1. “using the B3LYP function and 6-34G (d,p) basis set” -  not 6-34G (d,p) but 6-31G (d,p)

Response: The reviewer’s suggestion is accepted, and the authors have replaced 6-34G (d,p) with 6-31G (d,p).

  1. Both “Co-crystal” and “cocrystal” is used in the main text. Please decide in favor of one of them.

Response: As per the reviewer’s suggestions, the authors have changed all the words in the manuscript to “co-crystal”. The changes have been made in Table 05 (Rows: 3rd and 5th, Column: 4th)

  1. There are some more relevant works that can be mentioned (doi 10.1039/D0CE01820A, etc.)

Response: As per the suggestion, the authors have added the said article in section 5.3 (reference no. 88 in the manuscript).

Reviewer 2 Report

The manuscript by Dhondale R. Madhukiran and co-authors, submitted to “Pharmaceutics”, reviews recent co-crystallization studies trying to overcome hygroscopicity in drug formulations.

The review is generally well written, focusing on recent and relevant studies about the title theme, although it could be more comprehensive, since it is focused on a limited number of references. Nevertheless, despite the somewhat limited scope, with a few dozens of references, I believe it is sufficient quality to warrant its publication in this journal, provided that the manuscript is revised.

I will now indicate a number of necessary revisions to the article:

Given the limited number of references cited by the authors for the use of co-crystals aiming to address hygroscopicity issues, I believe the 8 self-citations (3, 8, 28, 34, 35, 46, 60, 105) is too much. The authors should make an effort to cite more references referring to co-crystalization and hygroscopicity an to limit the number of self-citations.

The review introduces the problems created by the inherent hygroscopicity of APIs and co-crystal synthesis as one of the possible ways to overcome those issues. Because sometimes the term is not very precisely used, the authors should provide the definition of “co-crystal”. This should clearly include supramolecular association of different substances in a common crystalline lattice with a precise stoichiometry, excluding solvates, co-amorphous mixtures, etc.

Also, other possible advantages of co-crystals beyond hygroscopicity improvement should also be cited. For instance, in the current version of the manuscript, solubility enhancement is mentioned for salts and not for co-crystals.

Letters symbolizing quantities should be italicized: for instance, Tg should be written as Tg.

Substance names are common nouns and therefore should not be capitalized (correct e.g. theophylline, berbetine, isosorbide, etc.).

“Co-crystals approach” is used several times in the text and in Figure 1. Alternative expressions like “co-crystallization approach”, as in the title, or simply “co-crystallization” would be grammatically preferable.

The sentence “Co-crystals provide an effective approach to enhancing the stability of APIs through multiple ways.” is not explanatory nor referenced. What “multiple ways” are these, and how cand they enhance the stability? Is this stability in a thermodynamical sense, or does it refer to permanence in the same solid form over shelf time?

The sentence “Co-crystals are almost very safe” is not clear. Change it into something more meaningful.

Figure 2 should be revised. Using absolute number of publications in B) when comparing with A) in not very helpful. I suggest using relative numbers in B) by dividing for numbers in A). Also, using only ScienceDirect, it takes data from a single publisher (Elsevier), which is not very representative. A more comprehensive database, such as Web of Science or Scopus, should be used.

The fragment “The possible structural and molecular mechanisms by which the co-crystals show

reduced hygroscopicity is still under exploration. Various researchers have constantly been trying to undermine this particular aspect of co-crystals that enhances their hygroscopic stability. Also, the choice of coformer for a particular API molecule depends on the property under question that needs to be enhanced.” is not well-written and needs to be revised and also to include proper references.

Given that “RH” was defined in the introduction as relative humidity, “equilibrium RH” does not make much sense, and “ERH is water activity can be expressed as a percentage” is not grammatically correct. Please define ERH unequivocally.

Figure 4 is cropped. Also, the respective CSD codes should be given in the caption.

Sections 3 and 4 of the manuscript are too extensive and deal mostly with drugs that are not co-crystals. They should be trimmed considerably and more focused on co-crystals.

Revise typo in last paragraph in the conclusions (“(…) molecular structures in co-crystals wh. We(...)”).

Author Response

The authors would like to thank the reviewer for the suggestions and for helping improve the manuscript's quality.

The manuscript by Dhondale R. Madhukiran and co-authors, submitted to “Pharmaceutics”, reviews recent co-crystallization studies trying to overcome hygroscopicity in drug formulations.

The review is generally well written, focusing on recent and relevant studies about the title theme, although it could be more comprehensive, since it is focused on a limited number of references. Nevertheless, despite the somewhat limited scope, with a few dozens of references, I believe it is sufficient quality to warrant its publication in this journal, provided that the manuscript is revised.

I will now indicate a number of necessary revisions to the article:

  1. Given the limited number of references cited by the authors for the use of co-crystals aiming to address hygroscopicity issues, I believe the 8 self-citations (3, 8, 28, 34, 35, 46, 60, 105) is too much. The authors should make an effort to cite more references referring to co-crystallization and hygroscopicity and to limit the number of self-citations.

Response: As per the suggestion by reviewer, the authors have removed two of the references (03 and 28) from the manuscript. The article of reference no. 46 does not belong to our group and is not removed. Also, few more works related to the co-crystals and hygroscopicity have been added/explained in text as given below:

Section 5.2- First paragraph, related to lattice energy calculations (reference no. 83).

Section 5.2- First paragraph, related to cocrystal stability (reference no. 84).

Section 5.3- Co-crystals of L-carnitine (reference no. 88).

Section 5.4- Fifth paragraph, explained in text the co-crystallization of Basella rubra extract with sucrose and gum-acacia to improve stability (reference no. 27).

Section 5.4- Fifth paragraph, related to co-crystal of phenazopyridine hydrochloride with phthalimide (reference no. 97).

  1. The review introduces the problems created by the inherent hygroscopicity of APIs and co-crystal synthesis as one of the possible ways to overcome those issues. Because sometimes the term is not very precisely used, the authors should provide the definition of “co-crystal”. This should clearly include supramolecular association of different substances in a common crystalline lattice with a precise stoichiometry, excluding solvates, co-amorphous mixtures, etc.

Response: The authors have included the definition of co-crystals in Page 05 Line 102 as per reviewer’s suggestion.

  1. Also, other possible advantages of co-crystals beyond hygroscopicity improvement should also be cited. For instance, in the current version of the manuscript, solubility enhancement is mentioned for salts and not for co-crystals.

Response: Other advantages of co-crystals is mentioned in the page 05 Line 99-100 as - “Also, a significant number of publications have shown research and development of co-crystals and their applications to improve aqueous solubility, dissolution rate, manufacturability, and oral bioavailability of API’s”. The solubility advantages of co-crystals have also been mentioned in Section 05, Page 18, Line 407-409. Considering the scope of article, the other advantages of co-crystals has not been discussed in detail.

  1. Letters symbolizing quantities should be italicized: for instance, Tg should be written as Tg.

Response: As per the suggestion from reviewer, letters symbolizing quantities have been italicized by the authors.

  1. Substance names are common nouns and therefore should not be capitalized (correct e.g. theophylline, berberine, isosorbide, etc.).

Response: The changes have been made wherever applicable (such as berberine, adiphenine, palmatine, theophylline, isosorbide, hydrochloride) in manuscript as per the suggestion.

  1. “Co-crystals approach” is used several times in the text and in Figure 1. Alternative expressions like “co-crystallization approach”, as in the title, or simply “co-crystallization” would be grammatically preferable.

Response: The authors have made changes in the following places

Page 03, Line 72 – Replaced “co-crystal approach” with “formation of co-crystals”

Page 05, Line 94 – Replaced “Co-crystal approach” with “co-crystallization of API”

  1. The sentence “Co-crystals provide an effective approach to enhancing the stability of APIs through multiple ways.” is not explanatory nor referenced. What “multiple ways” are these, and how can they enhance the stability? Is this stability in a thermodynamical sense, or does it refer to permanence in the same solid form over shelf time?

Response: The authors would like to humbly inform that the mechanisms by which hygroscopicity is reduced after co-crystal formation is explained in Section 5 of the review article. To provide clarity in the sentence, the phrase “which is discussed in detail in the later section of the review article” has been added in line 88 of page 05. Also, the co-crystal exhibit improved thermodynamic stability as well as chemical stability (shelf-life).

  1. The sentence “Co-crystals are almost very safe” is not clear. Change it into something more meaningful.

Response: The sentence of line 89, page 05 has been rephrased as “The use of substances from the USFDA list of generally recognized as safe (GRAS) and nutraceuticals as cofomers makes the cocrystals devoid of any toxicity due to the use of coformers.” and in support have added reference no. 29.

Thakuria et al. https://doi.org/10.3390/cryst8020101 (reference no. 29)

  1. Figure 2 should be revised. Using absolute number of publications in B) when comparing with A) in not very helpful. I suggest using relative numbers in B) by dividing for numbers in A). Also, using only ScienceDirect, it takes data from a single publisher (Elsevier), which is not very representative. A more comprehensive database, such as Web of Science or Scopus, should be used.

Response: As per the reviewer’s suggestion for figure 2, the authors have now changed the graph and used relative number of publications in B). Also, the cumulative number of publications every 4 years has been plotted for both A) and B). The start year has been changed from 2002 to 2003, both, in text (page 05, line 92) and in the figure 2 caption. The publication trends from other databases such as Web of Science and Scopus is also included as a supplementary file of the review article.

  1. The fragment “The possible structural and molecular mechanisms by which the co-crystals show reduced hygroscopicity is still under exploration. Various researchers have constantly been trying to undermine this particular aspect of co-crystals that enhances their hygroscopic stability. Also, the choice of coformer for a particular API molecule depends on the property under question that needs to be enhanced.” is not well-written and needs to be revised and also to include proper references.

Response: The above paragraph is introducing the problem statement of the review article i.e., the mechanism of co-crystals in solving hygroscopicity issues of APIs and the same has been explained in detail in the section 5 of the review article.

  1. Given that “RH” was defined in the introduction as relative humidity, “equilibrium RH” does not make much sense, and “ERH is water activity can be expressed as a percentage” is not grammatically correct. Please define ERH unequivocally.

Response: After thorough deliberation, the authors have come to conclusion that the part of the paragraph is creating confusion and can be removed from the manuscript, and hence it was decided to delete the part from the manuscript to ensure clarity.

  1. Figure 4 is cropped. Also, the respective CSD codes should be given in the caption.

Response: Figure 4 is not cropped, but an alignment issue. Now, the authors have modified the figure and will be visible on a single page. Also, the identifiers codes have been mentioned in the caption of the figure 4 and the codes are mentioned in the revised manuscript and below as well for the ready reference of the reviewer.

Compound

CCDC Identifier Number

Berberine chloride-tetrahydrate

1306671

Berberine chloride-citric acid co-crystal

1857453

Berbeine chloride-2emodin-ethanol co-crystal

1862517

Palmatine saccharinate salt

1977091

Palmatine sulfosalicyate salt

2053643

Palmatine-gallic acid cocrystal

2075058

  1. Sections 3 and 4 of the manuscript are too extensive and deal mostly with drugs that are not co-crystals. They should be trimmed considerably and more focused on co-crystals.

Response: In these two sections, the authors have focused on the hygroscopic APIs that are already in the market and have tried to summarize the problems associated in handling of hygroscopic APIs. However, as per the suggestion, the authors have trimmed the section 3 and 4 as much as possible. Few of the editions are mentioned below:

  • In Line 231 of page 13, replaced “the concentration of water in the final product causes the degradation of drugs through hydrolytic reactions” with “The water may degrade APIs by inducing hydrolytic reactions”.
  • Line 243 of page 14, removed “like inlet air RH, spray rate, etc.
  • Line 248 of page 14, removed sentence “the lowest crystallinity cellulose grade presented the highest stability compared to cellulose of higher crystallinity”.
  • Line 262 of page 14, removed “and its distribution”.
  • line 265-266, removed “The stability of the liquid bridges decides the boundary between the two forms of water retention which occurs at a particular RH” from.
  • Line 269, page 14, removed “by computing Carr’s index, Hausner’s ratio, and angle of repose”.
  • Line 278-283, removed complete paragraph in page 15 and added the sentence “Also, the flow properties of hygroscopic drug, theophylline is affected to greater extent due to water sorption.”
  • Line 288-289, page 15, Removed “Lubrication of the die wall occurs due to a reduction in density variation, which permits the transfer of the applied force to be transmitted to the lower punch through the compact”.
  • Line 322, page 16, removed “Disintegrants can absorb moisture during storage which affect the product’s shelf life”.
  • Line 366, page 18, removed “Bound water is now free to take part in many reactions like hydrolysis, leads to degradation of moisture-sensitive drugs and excipients”.
  • Line 370, page 18, removed “Therefore, hygroscopic excipients should not be combined with moisture-sensitive drugs.
  • Line 374, page 18, removed “Therefore, a wet granulation process must be avoded for moisture-sensitive drugs to prevent any degradation”.
  • Line 376-378, page 18, replaced “If the dose quantity is less than 0.5mg, roller compaction or dry granulation is unable to provide content-uniformity. Under such conditions” with “for low-dose drugs (< 0.5mg)”.
  • Line 383-385, page 19, replaced “which further complicates the tablet manufacturing process of hygroscopic drugs. Such cases require sampling and testing to ensure batch inconsistencies which ultimately increases the cost of manufacturing the hygroscopic drugs” with “thus, causing inconsistencies and require extensive sampling which increase cost of manufacturing of hygroscopic drugs”.
  • Line 387, Page19, replaced “moisture content was below 5% RH” with “RH was <5%”.
  • Line 393, Page 19, removed “The production facility requires specialized design for the manufacturing of hygroscopic drugs”.
  1. Revise typo in last paragraph in the conclusions (“(…) molecular structures in co-crystals wh. We(...)”).

Response: Corrected the typo, and removed “wh”.

Reviewer 3 Report

In this paper, mainly the effect of co-crystallization on hygroscopicity to enhance the stability of moisture-sensitive drugs is reviewed. 

 I understand the effect of co-crystal formation on hygroscopicity.

 However, Reviewer has the following questions due to the effect on solubility.

In Table1 and Table3, various hygroscopic data are compared between co-crystals or different salt forms. Some points are not clear. It is shown that by forming of co-crystal the reduction of solubility is noted in some cases. What is the reason about this?  How do you think about the change of solubility of raw material by the conformation of co-crystal.

In the case of the formation of salts the solubility seems to decrease significantly.  What is the reason for the different effect on solubility between cocrystal and salts.

I hope more brief description for the forcussed points in overall manuscript..

Author Response

The authors would like to thank the reviewer for the suggestions and for helping improve the manuscript's quality.

This paper reviews the effect of co-crystallization on hygroscopicity to enhance the stability of moisture-sensitive drugs. I understand the effect of co-crystal formation on hygroscopicity. However, the Reviewer has the following questions due to the effect on solubility.

  1. In Table 1 and Table 3, various hygroscopic data are compared between co-crystals or different salt forms. Some points are not clear. It is shown that by forming of co-crystal the reduction of solubility is noted in some cases. What is the reason about this?  How do you think about the change of solubility of raw material by the conformation of co-crystal.

Response: The solubility of the co-crystal form might reduce due to the usage of a hydrophobic coformer that increases the overall hydrophobicity of the co-crystal and cause a reduction in its aqueous solubility. This is explained by citing the examples of berberine chloride-emodin cocrystal (ref. 40) and oxymatrine-hydroxy 2-naphthoic acid cocrystal (ref. 36). In case of palmatine chloride-gallic acid co-crystals, the authors have not mentioned the exact reason for reduced solubility after co-crystal formation but have suggested that the co-crystal form has reduced ability to bind with external water molecules (ref. 23).

References: Reference no. 36- https://doi:10.1021/acs.cgd.1c00205 (Qi et al.)

        Reference no. 40- https://doi.org/10.1021/acs.cgd.8b01257 (Deng et al.)

        Reference no. 23- https://doi.org/10.1016/j.jddst.2021.102839 (Zhang et al.)

  1. In the case of the formation of salts the solubility seems to decrease significantly.  What is the reason for the different effect on solubility between cocrystal and salts.

Response: Authors would like to confirm that not all salt forms show reduced aqueous solubility. In the manuscript, the comparison is drawn between the hydrochloride/chloride salt form with other salts such as sulfosalicyate, citrate, oxalate, etc. (Table 03 and section 2.2). The reduction in solubility of other salt forms compared to chloride salts is due to an increase in the carbon length in the counter ion that increases the hydrophobicity and also due to the reduced dissociation of other salt forms, and as a result, ionization of drug is impacted, and solubility reduces (ref. 15).

Reference: Reference no. 15 - https://doi.org/10.1021/acs.cgd.2c00025 (Rebeiro et al)

  1. I hope the more brief description of the focused points in the overall manuscript.

Response: Authors have addressed all the queries raised by the reviewer, along with references.

Reviewer 4 Report

The review article by Madhukiran et al. provides a timely overview of co-crystallization in the context of stability moisture-sensitive drugs. Although co-crystallization has received significant attention in the past decade as a technology to improve the physicochemical properties of active pharmaceutical ingredients (APIs), the literature concerning the cocrystallization of hygroscopic forms has been sparse and hard to find. The authors have provided a comprehensive introduction to the challenges of hygroscopic APIs and current strategies to improve their stability, including crystal engineering techniques (e.g., polymorph control, cocrystallization) with representative references for each case.

Overall, the review paper is well-written, illustrative, and motivates the community to explore cocrystallization to solve relevant problems concerning the stability of hygroscopic APIs. Cocrystallization is of interest to the community from a fundamental level because of its numerous applications to pharmaceutics. Because of the abovementioned, I recommend the paper for publication in Pharmaceutics after carefully evaluating the following suggestions.

1.- Authors have highlighted some of the important intermolecular/supramolecular interactions and synthons (e.g., 5.1. Molecular Orientation and Aromatic Interactions) involved in cocrystal formation. However, the toolbox of supramolecular interactions is vast. A separate section that includes other common interactions involved in hygroscopic APIs (e.g., types of hydrogen bonds, halogen bonds (if any), homosynthons, heterosynthons) would be useful for the reader.

2.- The schematics and figures (e.g., Figures 3, 4) were illustrative to understand the concept of cocrystallization. It would be informative to add an additional figure in section 5.4 (Recent Works) to illustrate recent cocrystallization approaches to improve APIs' stabilities.

3.- It was not entirely clear to me the approach mentioned in the first paragraph of section 5.2. Specifically, how “The triclinic form of the malonic acid-theophylline co-crystal showed a minimal lattice energy compared to monoclinic malonic acid-theophylline with anhydrous theophylline having the highest lattice energy. This also explains the comparatively higher stability of triclinic form of co-crystal against hydration”. Please elaborate further, with specific values, if needed.

Author Response

The authors would like to thank the reviewer for the suggestions and for helping improve the manuscript's quality.

The review article by Madhukiran et al. provides a timely overview of co-crystallization in the context of stability moisture-sensitive drugs. Although co-crystallization has received significant attention in the past decade as a technology to improve the physicochemical properties of active pharmaceutical ingredients (APIs), the literature concerning the cocrystallization of hygroscopic forms has been sparse and hard to find. The authors have provided a comprehensive introduction to the challenges of hygroscopic APIs and current strategies to improve their stability, including crystal engineering techniques (e.g., polymorph control, cocrystallization) with representative references for each case.

Overall, the review paper is well-written, illustrative, and motivates the community to explore cocrystallization to solve relevant problems concerning the stability of hygroscopic APIs. Cocrystallization is of interest to the community from a fundamental level because of its numerous applications to pharmaceutics. Because of the abovementioned, I recommend the paper for publication in Pharmaceutics after carefully evaluating the following suggestions.

  1. Authors have highlighted some of the important intermolecular/supramolecular interactions and synthons (e.g., 5.1. Molecular Orientation and Aromatic Interactions) involved in cocrystal formation. However, the toolbox of supramolecular interactions is vast. A separate section that includes other common interactions involved in hygroscopic APIs (e.g., types of hydrogen bonds, halogen bonds (if any), homosynthons, heterosynthons) would be useful for the reader.

Response: As per the suggestion by the reviewer, the authors have added a paragraph briefly describing the types of hydrogen bonding interactions possible in the hygroscopic substances in Section 2.1, lines 145-151.

  1. The schematics and figures (e.g., Figures 3, 4) were illustrative to understand the concept of cocrystallization. It would be informative to add an additional figure in section 5.4 (Recent Works) to illustrate recent cocrystallization approaches to improve APIs' stabilities.

Response: As per the suggestion, the authors have added an additional illustration (Figure 11) in section 5.4.

  1. It was not entirely clear to me the approach mentioned in the first paragraph of section 5.2. Specifically, how “The triclinic form of the malonic acid-theophylline co-crystal showed a minimal lattice energy compared to monoclinic malonic acid-theophylline with anhydrous theophylline having the highest lattice energy. This also explains the comparatively higher stability of triclinic form of co-crystal against hydration”. Please elaborate further, with specific values, if needed.

Response: There were some unclear sentences in the first paragraph of section 5.2, and the authors have now rectified the same and has been changed to “The co-crystal having lower lattice energy is known to have reduced hygroscopicity when compared to the co-crystal of higher lattice energy [84]. Lattice energy calculations can also be used to predict the stability of the co-crystals. The lattice energy of the triclinic and monoclinic form of theophylline-malonic acid co-crystals are -231.7 kJ/mol and -267.6 kJ/mol, respectively. This infers that the monoclinic form is more stable compared to the triclinic form of theophylline-malonic acid co-crystal, which can be regarded as the metastable form [85]”.

References:      https://doi.org/10.1016/j.ijpharm.2021.120537 (Reference no. 84)

                              https://doi.org/10.1039/D1RA08389A (Reference no. 85)

Reviewer 5 Report

In this review "Co-crystallization approach to enhance the stability of mois-ture-sensitive drugs" authors well explain and document the approaches that have taken place in improving the stability of API. They have clearly mentined the objective and presented in a clear manner so that the readers could understand well. Usage of clear graphical presentations also improves the clarity of the paper. I believe this will be a very useful review for the readers as there is an increase of research going on in the field of crystal engineering and co-crystal techniques. I highly recommend this paper for publication.

I also would like authors to consider the following before final submission,

1. Page 11 Figure 4 , full image is not visible

2. Figure 7- it is off the page, can reduce the size

3. text and figures are not centered properly, text is more center aligned and it avoids the clarity of the paper.

4. Some paragraphed are justified and some are not, need the same format throughout the paper. 

Author Response

The authors would like to thank the reviewer for the suggestions and for helping improve the manuscript's quality.

In this review "Co-crystallization approach to enhance the stability of mois-ture-sensitive drugs" authors well explain and document the approaches that have taken place in improving the stability of API. They have clearly mentioned the objective and presented in a clear manner so that the readers could understand well. Usage of clear graphical presentations also improves the clarity of the paper. I believe this will be a very useful review for the readers as there is an increase of research going on in the field of crystal engineering and co-crystal techniques. I highly recommend this paper for publication.

I also would like authors to consider the following before final submission,

  1. Page 11 Figure 4, the full image is not visible

Response: Due to the alignment issue, Figure 4 was not fully visible. Now, the authors have rectified the issue, and the figure will be visible on a single page.

  1. Figure 7- it is off the page, can reduce the size

Response: The size of figure 7 is reduced as per suggestion.

  1. Text and figures are not centered properly, text is more center aligned and it avoids the clarity of the paper.

Response: Following the reviewer’s, the authors have aligned the figures and texts to improve clarity.

  1. Some paragraphed are justified and some are not, need the same format throughout the paper. 

Response: As per the suggestion from the reviewer, the authors have justified all the paragraphs in the manuscript.

Round 2

Reviewer 2 Report

I am now satisfied with the authors revisions.